# Geodesic Self-Attention for 3D Point Clouds

**Zhengyu Li**[1] **Xuan Tang**[1] **Zihao Xu**[1] **Xihao Wang**[2] **Hui Yu**[3]

**Mingsong Chen**[1] **Xian Wei**[1]*

[1]East China Normal University
[2]Technical University of Munich
[3]FJIRSM, Chinese Academy of Sciences
xwei@sei.ecnu.edu.cn

## Abstract

Due to the outstanding competence in capturing long-range relationships, self-attention mechanism has achieved remarkable progress in point cloud tasks. Nevertheless, point cloud object often has complex non-Euclidean spatial structures, with the behavior changing dynamically and unpredictably. Most current self-attention modules highly rely on the dot product multiplication in Euclidean space, which cannot capture internal non-Euclidean structures of point cloud objects, especially the long-range relationships along the curve of the implicit manifold surface represented by point cloud objects. To address this problem, in this paper, we introduce a novel metric on the Riemannian manifold to capture the long-range geometrical dependencies of point cloud objects to replace traditional self-attention modules, namely, the **G**eodesic **S**elf-**A**ttention (GSA) module. Our approach achieves state-of-the-art performance compared to point cloud Transformers [13, 10, 44, 26] on object classification, few-shot classification and part segmentation benchmarks.

## 1 Introduction

Motivated by the rapid development of applications in robotics, autonomous vehicles, and so on, research in 3D information processing with **D**eep **N**eural **N**etwork**s** (DNNs) is likewise undergoing a revolution. Point cloud has garnered considerable research attention as one of the most readily available and vital data in the 3D vision field. The essence of a point cloud is discrete 3D coordinates sampled from a continuous 3D shape, presenting a non-Euclidean property. Therefore, it is typically considered to represent a geometric surface or a 2D manifold embedded in Euclidean space. In addition, 3D point cloud data mainly consists of geometric coordinate information and is generally processed only with it, whereas word vectors and image pixel matrices contain rich semantic features. The geometric features should be the first concern when analyzing point clouds. Although there have been

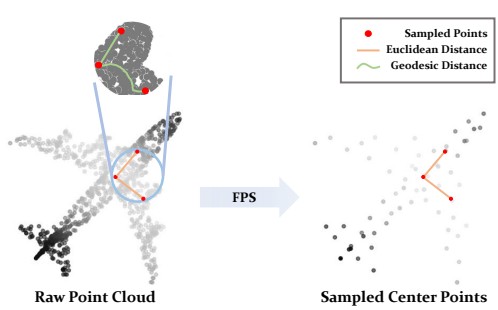

Figure 1: A typical geometric semantics confusion scenario, where two pairs of points (upper-middle and lower-middle pairs) on raw point cloud have similar Euclidean distances but get different geodesic distances and semantics.

---

*Corresponding author

36th Conference on Neural Information Processing Systems (NeurIPS 2022).

a lot of impressive works on point cloud geometric analysis methods in the past [28, 38, 15, 13], most of them are based on the Euclidean or local geometry, which is not sufficient to represent the non-Euclidean geometric semantics of point clouds accurately. One of the most typical geometric semantics confusion scenarios is illustrated in Fig. 1 to further elaborate on this assumption. To address this problem, we hope to introduce an accurate non-Euclidean metric that can capture both the local and global features of the point cloud. Associating the powerful capability of capturing the long-range dependencies of Transformer [36], it could be the critical bridge to introduce such a metric.

Nowadays, Transformer is another popular research topic. Relying on the powerful ability to capture global information and long-range dependencies aggregation of self-attention mechanism, various types of networks with Transformer as the backbone have dominated in areas such as Natural Language Processing (NLP)[7, 30, 3] and Computer Vision (CV) [9, 2, 24, 14, 11]. Although existing dot-product self-attention [9] is effective enough in computer vision, or to be specific, in image processing, in 3D tasks, especially in point cloud tasks, we infer that the dot product multiplication in Euclidean space may not be the optimal attention solution, due to its limitation in capturing internal non-Euclidean structures of point cloud objects. Since existing works [13, 10, 44, 26] have achieved impressive results with dot product self-attention, we do not want to challenge their contributions. Exploring whether there are any better and geometrically relevant metrics for the self-attention mechanism on point cloud tasks is the goal of the work.

Inspired by Riemannian geometry [17], we introduce a metric on the Riemannian manifold into point cloud analysis. For a Riemannian manifold, the geodesic generally represents the shortest path between two points along a manifold surface and dictates the geometric relationship. From a mathematical perspective, the geodesic describes the locally distance-minimizing curve, which can obtain a more accurate relationship measurement than the metric in Euclidean space (commonly considered as a straight line) for the implicit surface features represented by point cloud feature vectors. In other words, the closer the geodesic distance between two data points, the more related they are. Therefore, we compute the attention map by point-wise geodesic distance rather than the dot product multiplication in Euclidean space, to capture the accurate global dependencies between point cloud data points.

In summary, in this paper, we propose a novel **G**eodesic **S**elf-**A**ttention (GSA) module in Transformers for point cloud tasks and introduce it into a standard Transformer structure, namely, **Point G**eodesic **T**ransformer (Point-GT). Our main contributions are as follows:

- To the best of our knowledge, we are the first to throw out the assumption that the metric on the Riemannian manifold should be more appropriate for computing an attention map for the point cloud Transformers than the metric on Euclidean space. From this assumption, we propose geodesic self-attention, which first utilizes geodesic to measure the topological attention relationships of the underlying geometric structure represented by the point cloud.

- To verify the feasibility of the assumption, we propose two geodesic distance computation methods for the point cloud features. We apply the geodesic self-attention to the standard Transformer structure, achieving better convergence and state-of-the-art performance with fewer parameters and faster computation.

## 2    Related Work

**Geometry Analysis of Point Cloud**    The point cloud is one of the most readily available 3D data in our daily lives, which can be discretely sampled from various 3D models or captured by 3D scanners (Lidar, RGB-D, etc.). Due to its irregular, non-grid characteristics, it was common to project the original point cloud onto a multi-view map [43, 20] or voxel [6, 31, 32] to utilize well-explored 2D convolution for processing in the early days. PointNet [27] pioneered a method of embedding point cloud coordinates directly using **M**ulti**L**ayer **P**erceptron (MLP), which inspired many subsequent point cloud analysis efforts. Based on PointNet, PointNet++ [28] introduced a hierarchical aggregation paradigm for point cloud local geometric information and has become a common point cloud embedding method in many exciting point cloud works nowadays. DGCNN [38] proposed a graph-based point cloud processing method which constructs local dynamic graphs to characterize edge relationships between base points and neighboring nodes and utilizes EdgeConv to learn implicit point cloud topological geometric features. Inspired by Transformers, PCT [13]

first combined the self-attention mechanism with edge features and proposed the Offset-Attention. Although the above works have all made significant contributions to the geometric analysis of point cloud data, most of the past geometric explorations of point clouds have only considered learning from the Euclidean metric or the local geometry of point clouds. Inspired by the global information capturing ability of the self-attention mechanism and the geometric representation ability of geodesic, we combine both to explore the global geometry of point clouds.

**Geodesic Distance Computation**    In previous work, the geodesic distance between 3D data is often computed by path wandering on the mesh surface [4, 5, 33] or vertex shortest path estimation [8, 12, 16]. On the one hand, these works have an unbearable time complexity (generally above $O(n^2)$ for one graph). On the other hand, most of the point cloud data in real-world scenarios do not contain meshes, making it even more challenging to obtain geodesic distances. In order to reduce the computational cost of geodesic distance during network training, GeoNet [15] approaches a **G**eodesic **M**atching (GM) layer to learn potential geodesic features. Before training the GM layer, the geodesic distance of sampled points on the mesh surface has been manually computed as ground truth. After training the GM layer, GeoNet utilizes the trained model to learn the potential geodesic features of the extracted point cloud features. Although the GM layer alleviates the difficulty of acquiring geodesic features during network training, we believe that acquiring explicit geodesic features tends to increase the inductive bias and reduce the learnability and robustness of the network. Besides, this matching method is more suitable for feature embedding and cannot be utilized to calculate the relative geodesic attention. Whiteley *et al*. [39] explore a graph-based method for recovering the geodesic distance of data features from a high-dimensional Hilbert space and theoretically prove the feasibility. Qi *et al*. [29] study few-shot learning with the Riemannian metric by projecting the features extracted from Euclidean space to **O**$blique$ **M**$anifold$ ($\mathcal{OM}$). Since the $\mathcal{OM}(n, m)$ is considered as a Riemannian submanifold of the embedding Euclidean space $\mathbb{R}^{n \times m}$ with unit Euclidean norm, which is an intrinsic property similar to the $l_2\text{-}norm$ in Euclidean space. We consider that such mapping methods can preserve the geometric features of the data and can be utilized for independent geometric analysis. Inspired by these works, we propose two methods from the graph and data manifold theory, respectively, to capture dynamic topological features of point cloud data and obtain geodesic attention maps.

**Point Cloud Transformers**    Recently, relying on the powerful sequential data processing and global features aggregation capability, Transformers have achieved great success in NLP and CV. 3D point clouds, a prevalent topic in computer vision, can also be considered special sequence data, which makes it naturally suitable to be processed by Transformers. In addition, to the potential future applications of multimodal work, it is vital to explore how to strengthen the performance of Transformers, a typical bridge of the multimodal data, in point cloud tasks. PCT [13] is one of the earliest works to introduce the Transformer structure to point cloud tasks. It designs a modified self-attention mechanism with the core idea of Edgeconv [38]. PoinTr [44] utilizes a modified DGCNN [38] as a feature embedding layer to capture local point cloud geometric information. Receiving inspiration from PoinTr, Point-BERT [45] designs a **d**iscrete **V**ariational **A**uto**E**ncoder (dVAE) for generating point tokens, successfully introducing BERT-style [7] pre-training to point cloud tasks. Point-MAE [26] further explores pre-training methods for point cloud Transformers. Inspired by **M**asked **A**uto**E**ncoders (MAE) [14] self-supervised learning method, it implements a point cloud task pre-training pipeline with an entirely standard Transformer structure. These works build a solid foundation for future work on point cloud tasks with Transformers.

Since most of the previous work process the point cloud data, representing the underlying geometric surface structure of 3D objects, with only the geometry information (3-dimensional Cartesian coordinates), the geometric properties are critical to point cloud analysis. However, the above works are mostly based on Euclidean dot-product self-attention, which cannot accurately capture the geometric structure of the point cloud. We propose geodesic self-attention based on Riemannian geometry to overcome this dilemma.

## 3   Point Geodesic Transformer

This section will detail our approach of introducing geodesic self-attention into the point cloud Transformers structure. First, we will describe how the Transformers structure can be applied to the

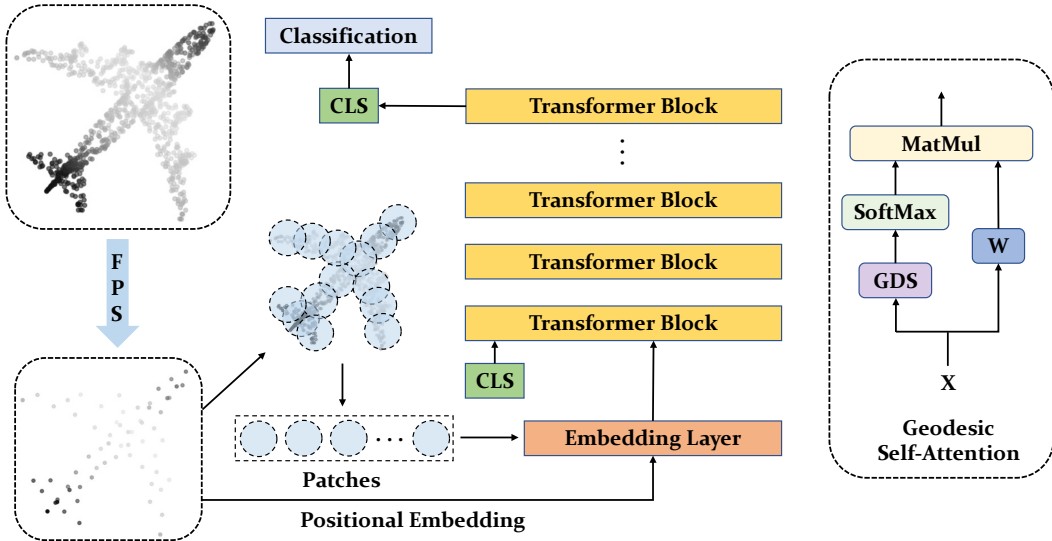

Figure 2: **The Point Geodesic Transformer (left) and the pipeline of the geodesic self-attention (right).** Taking the classification task as an example, we present the overall structure of Point-GT.

point cloud task. Then, we will describe how geodesic self-attention works in our network structure. The overall structure of the Point Geodesic Transformer is illustrated in Fig. 2.

## 3.1 Transformer Backbone

In order to compare with dot-product self-attention directly, inspired by Point-BERT [45], our main experiments are performed on the standard Transformer [36] structure, except for the modification of the self-attention function.

**Patches Generation** Unlike the classical visual tasks with regular grids, point clouds consist of unordered points describing 3D coordinates. It is challenging to find a suitable method to embed the point cloud feature. Following the previous point cloud Transformers [13, 10, 44, 26], we adopt an embedding strategy that encodes point clouds as irregular local point patches. To be specific, given an input point cloud $P \in \mathbb{R}^{N \times 3}$ ($P = \{p_1, p_2, p_3, \cdots, p_N\}$), we first perform **F**arthest **P**oint **S**ampling (FPS) to sample $g$ center points from the raw point cloud data. Then, the $k$-**N**earest **N**eighbor ($k$-NN) algorithm captures the $g$ centers' $k$ nearest neighbor points, and a normalization operation is performed to make the local neighbors unbiased.

**Points Embedding** In order to maintain the permutation invariance of the point cloud [27], we do not choose the patches embedding method like ViT [9], which simply flattens the input patches and embeds them with a trainable linear projection. For each input point patch, several MLP layers and aggregation operations which can be considered as a lightweight PointNet[27] are utilized to project the point patch to point embedded feature. Given the center point $x$ and its normalized neighbors $\mathbf{X} = \{x_1, x_2, x_3, \cdots, x_k\}$, the embedding operations can be described as:

$$\mathbf{X}' = \oplus f_\Theta(\mathbf{X}), \tag{1}$$

where $\oplus$ denotes an aggregation function, $f_\Theta(\cdot)$ can be considered as an embedding layer and $\mathbf{X}'$ is the embedded point cloud feature. In this step, there are many alternative aggregation functions [38], such as $max\text{-}pooling$ and $mean\text{-}pooling$, and the aggregation function captures the local geometric information of the input point patch $\mathbf{X}$. To be consistent with the previous work [13, 10, 44, 26], we choose $max\text{-}pooling$ as the aggregation function here.

As for the positional embedding, a previous work[25] claims it is unnecessary to encode the position of the 3D point cloud feature for its raw data contains the $xyz$ coordinates. However, in our implementation, the normalization operation mentioned in the patches generation stage decouples the

relative positional relationships, so we need to map the positional information of the corresponding patch center points through a simple MLP to the patch features. Since the embedded feature can be processed directly by the standard Transformer, the structure of Transformer blocks is almost the same as the standard Transformer[36].

## 3.2 Geodesic Self-Attention

The original definition of attention function can be described as mapping a query and a set of key-value pairs to an output. For the self-attention function, the query and the key are the input vector itself [36]. Following the exact definition, the geodesic self-attention score map can be described as the point-wise geodesic distance scores of the input vectors. Although the 3D point cloud data represents the underlying shape of 3D objects, it does not contain an explicit manifold surface. Therefore, how to define the geodesic of the points and how to obtain the geodesic distance score are crucial. Inspired by previous work [39, 29], we propose two methods to compute the geodesic distance score based on graph and data manifold, respectively.

### 3.2.1 Geodesic Distance Score

**Graph-based Geodesic**   Whiteley *et al.* [39] recover the distances and coordinates between features through matrix decomposition and dimension reduction. Isomap [34] constructs the graph structure between feature points and computes the geodesic distances between nodes through $k$-NN and shortest path algorithms. Since the previous studies [38, 19] have supported the analysis of point clouds as a graph structure, we combine the thoughts from the above works, proposing a graph-based geodesic computation method. First, we construct the dynamic weighted adjacency graph for the input features of each Transformer block through $k$-NN. We believe that dynamically updating the adjacency graph can not only learn explicit geodesic features but also recover the latent topological features of point clouds, which enrich the representation power of point clouds [38]. During constructing the dynamic adjacency graph, we set the weights of the adjacency graph to Euclidean distance for adjacent points and infinity for non-adjacent points. Following the assumption that a manifold locally resembles Euclidean space, we apply Floyd's shortest path algorithm [12] to estimate the geodesic distance of all pairs of the points. Finally, the output is the **G**eodesic **D**istance **S**core (GDS) matrix.

---

**Algorithm 1** Graph-based Geodesic Distance Score

**Input:** $\mathbf{X} = \{x_1, x_2, x_3, \cdots, x_g\}$
**Output:** $GDS$
 1: $K = \lfloor \sqrt{g} \rfloor$                                                                  ▷ Set the threshold of neighbors.
 2: Construct neighborhood graph $G$ by connecting points i and j:
 3: **if** $i$ is one of the $K$ nearest neighbors of $j$ **then**
 4:     Set edge length equal to $d_{\mathbf{X}}(i, j)$
 5: **else**
 6:     Set edge length equal to $\infty$
 7: **end if**
 8: Compute $GDS$ through Floyd's shortest paths algorithm.
 9: **return** $GDS$

---

Since Floyd's algorithm is nonlinear and cannot be computed in parallel, we use Numba [18] to implement a **C**ompute **U**nified **D**evice **A**rchitecture (CUDA) operator for computational acceleration. Each computation of the adjacency graph is implemented by a single CUDA kernel and achieves more than a hundred times faster computation.

**Data Manifold-based Geodesic**   Riemannian geometry is the study of smooth manifolds by introducing Riemannian metrics based on differential geometry, defining geometric concepts such as angles, length of curves, etc. The **O**$blique$ **M**$anifold$ ($\mathcal{OM}$) is a submanifold of the embedding Euclidean space with unit Euclidean norm columns and is always utilized for independent component analysis [1]. Formally, $\mathcal{OM}(n, g)$ is defined as:

$$\mathcal{OM}(n, g) = \left\{ \boldsymbol{P} \in \mathbb{R}^{n \times g} : \mathrm{diag}\left(\boldsymbol{P}^T \boldsymbol{P}\right) = \boldsymbol{I_g} \right\}, \tag{2}$$

where $diag(\cdot)$ denotes the diagonal matrix.

Previous work [29] models few-shot learning on $\mathcal{OM}$. The authors project the data features extracted in the Euclidean space onto $\mathcal{OM}$ and then design an oblique distance-based classifier to achieve the classification task for the projected features. Since the $\mathcal{OM}$ offers a unit $l_2$-$norm$ intrinsic property, we consider the above projection preserves the relative geometric structure of the features, which satisfies the need for the attention mechanism to capture relative dependencies. Following this notion, we constrain the point cloud features from Euclidean space to *oblique manifold* to extract the underlying geometric structure. Since the positional embedding recovers the relative geometric distance of the point cloud patches, we directly project the input patches embedded in Euclidean space onto $\mathcal{OM}$ to compute the implicit geodesic relations between patches. Considering the patch matrix $\boldsymbol{P} = \{p_1, p_2, \cdots, p_g\}$ does not naturally satisfy the Eq. 2, in other words, $\boldsymbol{P}$ is not a member of $\mathcal{OM}$. We apply a projection $Proj(\cdot)$ to get the manifold-valued point cloud feature matrix:

$$\boldsymbol{P} := \mathrm{Proj}\left(\boldsymbol{P}\right) = \mathrm{Cat}\left(\left\{\frac{p_i}{\|p_i\|}\right\}_{i=1}^{g}\right), \tag{3}$$

where $Cat(\cdot)$ denotes concatenate function and $\|\cdot\|$ denotes the square Frobenius norm in the ambient space.

After the projection, the geodesic distance of an input pair of points $\{\boldsymbol{Q}, \boldsymbol{K}\}$ on $\mathcal{OM}$ can be calculated as:

$$\mathrm{dist}(\boldsymbol{Q}, \boldsymbol{K}) = \sqrt{\sum_{i=1}^{n} \mathrm{arccos}^2\left(\mathrm{diag}\left(\boldsymbol{Q}^T\boldsymbol{K}\right)\right)_i}. \tag{4}$$

After solving the dilemma of obtaining the geodesic, we need to consider how the geodesic distance score guides our attention mechanism. Based on the principle that the more distant the geometric relationship is, the fewer the dependencies between the points, we simply make the geodesic distance score negative and apply a $softmax$ function to obtain the attention map weights. In addition, we have also tried to take the inverse of the geodesic distance score or normalize it before $softmax$. However, through the experiments, we found that the most naive process achieved the best result. We inferred that the redundant operation might impair the representation ability of GSA, for the distance between points is relatively small after the scaling of data augmentation. Overall, given the input $X$, the computation pipeline of GSA is illustrated in Fig. 2. Note that we do not implement linear transformations to project the input tensor $X$ like the standard dot product multiplication attention [36] in our geodesic distance score computation stage, because we find it might do harm to geometric structure feature extraction. Without linear transformations, our manifold-based GSA achieves faster implementation with fewer parameters. More details are illustrated in Section 4.4.

## 4 Experiments

In this section, we comprehensively evaluate the performance of Point-GT on several benchmarks, including the classification task, part segmentation and few-shot classification. In the ablation studies, we qualitatively and quantitatively assess the effectiveness of GSA. Some additional experiments are illustrated in supplementary materials.

### 4.1 Object Classification Tasks

**Object Classification on ModelNet40**   ModelNet40 [40], one of the most popular 3D object classification datasets, contains 12,311 synthesized CAD models from 40 categories. We split the dataset into 9,843 training and 2,468 validation instances following the previous standard practice. Standard random scaling and random translation are applied for data augmentation during the training. More experiment details are provided in supplementary materials.

The experiment results are presented in Table 1, and we denote our methods as Point-GT and Point-GT-MAE. The former is trained from scratch, and the latter is pre-trained with MAE [26] strategy[2]. Furthermore, we denote data manifold-based and graph-based methods with the suffixes

---

[2]We have tried to use BERT [45] strategy for pre-training, but the pre-trained network did not converge to a reasonable $loss$ value during pre-training. We also applied the BERT pre-training strategy on other point cloud Transformers and found a similar phenomenon, so we ruled out that the GSA should be responsible for this error and only evaluated the performance of the pre-trained Point-GT with MAE [26] strategy.

"-DM" and "-G", respectively. For a fair comparison, our methods utilize the standard voting method[23] during testing, and the different number of input points is also presented. Yu *et al.* [45] observe that increasing the number of input points cannot improve the performance of a model without pre-training. Therefore, we test Point-GT with 1,024 input points. Since the graph-based GSA still requires unaffordable computational overhead even after being accelerated with the CUDA operator, all the results of our Point-GT-MAE represented below are implemented with the data manifold-based GSA. When training from scratch, graph-based Point-GT achieves 92.6% accuracy without voting, which is enough to demonstrate its feasibility and superiority.

We compare our approach with several Transformer-based models. Among them, [T] denotes the Transformer model with some special designs of Transformer blocks or FeedForward Network [36], [ST] denotes a standard Transformer architecture, and [STP] represents a [ST] model with pre-training. Compared with the point cloud Transformers with dot-product self-attention, our Point-GT achieves an excellent result (93.3%) which is even better than a model, Point-BERT (93.2%), with pre-training when the input points are 1,024. Increasing the number of input points to 8,192, we can also find that Point-GT with MAE pre-training achieves state-of-the-art performance (94.1%) and is better than Point-MAE (94.0%), which is the best Transformer-based point cloud work up to date. Compared with the point cloud Transformers with special self-attention, we test the GSA on PCT [13] by modifying its self-attention function. Considering dot-product score plays a relatively less important role in PCT, our GSA does not bring as much of a boost as it does in vanilla Transformer. However, a minor improvement is also valuable. We also designed an experiment on the rotation robustness, and the results are presented in supplementary materials.

Table 1: **Classification results on ModelNet40.** We report the accuracy (%) and the number of sampled points in the input.

| Methods | #Point | Acc. |
|---|---|---|
| PointNet [27] | 1k | 89.2 |
| PointNet++ [28] | 1k | 90.5 |
| PointCNN [21] | 1k | 92.2 |
| DGCNN [38] | 1k | 92.9 |
| RS-CNN [23] | 1k | 92.9 |
| [T] PCT [13] | 1k | 93.2 |
| [T] NPCT [13] | 1k | 91.0 |
| [T] PCT-GSA-DM (Ours) | 1k | 93.3 |
| [ST] Transformer [45] | 1k | 91.4 |
| [ST] Point Transformer [10] | 1k | 92.8 |
| [ST] Point-GT-DM (Ours) | 1k | 93.3 |
| [STP] Point-BERT [45] | 1k | 93.2 |
| [STP] Point-MAE [26] | 1k | 93.8 |
| [STP] Point-GT-MAE (Ours) | 1k | 93.6 |
| [STP] Point-BERT [45] | 8k | 93.8 |
| [STP] Point-MAE [26] | 8k | 94.0 |
| [STP] Point-GT-MAE (Ours) | 8k | **94.1** |

**Object Classification on ScanObjectNN**   Since ModelNet40 is a noise-free artificial dataset and has been extensively studied, it has gradually failed to meet current research needs, so we also evaluate our approach on the ScanObjectNN [35] benchmark. ScanObjectNN contains about 15,000 objects sampled from 2,902 real-world instances with 15 categories. Since the real-world dataset will inevitably be affected by noise or occlusions, it is a much more challenging dataset for point cloud analysis methods. We follow previous works [45, 26] to evaluate our approach on three variants: OBJ-BG, OBJ-ONLY, and PB-T50-RS. In the comparison, our Point-GT highly improves the baseline Transformers on three variants without voting, and our Point-GT-MAE also outperforms the pre-trained point cloud Transformers.

## 4.2   Few-Shot Learning

Following previous works [45, 26], we conduct few-shot learning experiments on ModelNet40. Adopting a "$K$-way $N$-shot" setting, where $K$ classes are randomly selected from the dataset and N denotes the number of instances randomly sampled for each class, we train the model on $K \times N$ samples (support set) and another $20 \times K$ instances (query set) are randomly sampled from $K$ classes for evaluation. The results with the setting of $K \in \{5, 10\}$ and $N \in \{10, 20\}$ are presented in Table 3. We conduct ten independent experiments for each setting and report the mean accuracy as well as the standard deviation over ten runs. According to the results, our approach achieves state-of-the-art performance.

Table 2: **Classification results on ScanObjectNN.** We report the accuracy (%) of three different settings.

| Methods | OBJ-BG | OBJ-ONLY | PB-T50-RS |
|---|---|---|---|
| PointNet [27] | 73.3 | 79.2 | 68.0 |
| SpiderCNN [41] | 77.1 | 79.5 | 73.7 |
| PointNet++ [28] | 82.3 | 84.3 | 77.9 |
| DGCNN [38] | 82.8 | 86.2 | 78.1 |
| PointCNN [21] | 86.1 | 85.5 | 78.5 |
| BGA-DGCNN [35] | - | - | 79.7 |
| BGA-PN++ [35] | - | - | 80.2 |
| Transformer [45] | 79.9 | 80.6 | 77.2 |
| Point-GT-G (Ours) | 86.6 | 88.3 | 80.2 |
| Point-GT-DM (Ours) | 87.6 | 88.6 | 81.7 |
| Point-BERT [45] | 87.4 | 88.1 | 83.1 |
| Point-MAE [26] | 90.0 | 88.3 | 85.2 |
| Point-GT-MAE (Ours) | **90.7** | **89.5** | **85.7** |

Table 3: **Few-shot classification results on ModelNet40.** We report the mean accuracy (%) with the standard deviation over 10 independent experiments.

| Methods | 5-way | | 10-way | |
|---|---|---|---|---|
| | 10-shot | 20-shot | 10-shot | 20-shot |
| DGCNN-rand [37] | 31.6±2.8 | 40.8±4.6 | 19.9±2.1 | 16.9±1.5 |
| Transformer-rand [45] | 87.8±5.2 | 93.3±4.3 | 84.6±5.5 | 89.4±6.3 |
| Point-GT-G-rand (Ours) | 95.4±2.5 | 96.7±2.4 | 90.2±5.4 | 93.0±4.4 |
| Point-GT-DM-rand (Ours) | 93.1±3.9 | 96.1±3.4 | 90.1±5.6 | 92.5±4.9 |
| Point-BERT [45] | 94.6±3.1 | 96.3±2.7 | 91.0±5.4 | 92.7±5.1 |
| Point-MAE [26] | 96.3±2.5 | 97.8±1.8 | 92.6±4.1 | 95.0±3.0 |
| Point-GT-MAE (Ours) | **96.3±2.3** | **98.2±1.5** | **92.8±4.6** | **95.1±3.6** |

## 4.3 Part Segmentation

Object part segmentation is a challenging task with a high requirement of model representation capability. It aims to predict a more fine-grained class label for each point. We evaluate our Point-GT on ShapeNetPart [42] dataset, containing $16,881$ instances of 16 categories. Following previous works [45, 26], we sample $2,048$ points as input for each instance and utilize a simple segmentation head without any propagating operation or DGCNN [38]. Specifically, we concatenate the learned features from the $4th$, $8th$ and $12th$ layer of Transformer blocks and then apply $mean\text{-}pooling$ and $max\text{-}pooling$ separately to obtain two global features. Then, we up-sample the concatenated features to $2,048$, which is the size of input points to obtain the predicting features.

The mean IoU across all categories and the mean IoU across all instances are illustrated in Table 4. Our training from scratch model, Point-GT, outperforms the pre-training model Point-BERT, which demonstrates the superior fine-grained representation capability of the GSA.

Table 4: **Part segmentation results on the ShapeNetPart dataset.** We report the mIoU$_C$ (%), the mIoU$_I$ (%), as well as the IoU (%) for each categories.

| Methods | mIoU$_C$ | mIoU$_I$ | aero | bag | cap | car | chair | earphone | guitar | knife | lamp | laptop | motor | mug | pistol | rocket | skateboard | table |
|---|---|---|---|---|---|---|---|---|---|---|---|---|---|---|---|---|---|---|
| PointNet [27] | 80.39 | 83.7 | 83.4 | 78.7 | 82.5 | 74.9 | 89.6 | 73.0 | 91.5 | 85.9 | 80.8 | 95.3 | 65.2 | 93.0 | 81.2 | 57.9 | 72.8 | 80.6 |
| PointNet++ [28] | 81.85 | 85.1 | 82.4 | 79.0 | 87.7 | 77.3 | 90.8 | 71.8 | 91.0 | 85.9 | 83.7 | 95.3 | 71.6 | 94.1 | 81.3 | 58.7 | 76.4 | 82.6 |
| DGCNN [38] | 82.33 | 85.2 | 84.0 | 83.4 | 86.7 | 77.8 | 90.6 | 74.7 | 91.2 | 87.5 | 82.8 | 95.7 | 66.3 | 94.9 | 81.1 | 63.5 | 74.5 | 82.6 |
| Transformer [45] | 83.42 | 85.1 | 82.9 | 85.4 | 87.7 | 78.8 | 90.5 | 80.8 | 91.1 | 87.7 | 85.3 | 95.6 | 73.9 | 94.9 | 83.5 | 61.2 | 74.9 | 80.6 |
| Point-BERT [45] | 84.11 | 85.6 | 84.3 | 84.8 | 88.0 | 79.8 | 91.0 | 81.7 | 91.6 | 87.9 | 85.2 | 95.6 | 75.6 | 94.7 | 84.3 | 63.4 | 76.3 | 81.5 |
| Point-GT-G (Ours) | 83.94 | **85.9** | 84.7 | 83.7 | 89.4 | 80.4 | 91.2 | 77.0 | 91.7 | 87.6 | 85.6 | 96.0 | 74.0 | 95.3 | 84.6 | 62.7 | 77.5 | 81.7 |
| Point-GT-DM (Ours) | **84.15** | 85.8 | 84.3 | 84.5 | 88.3 | 80.9 | 91.4 | 78.1 | 92.1 | 88.5 | 85.3 | 95.9 | 77.1 | 95.1 | 84.7 | 63.3 | 75.6 | 81.4 |

## 4.4 Ablation Study

In this subsection, we conduct several controlled experiments to explore how to design our geodesic self-attention and compare the performance of different attention types. The experiments are based on the classification task on ModelNet40 and the number of input points is 1,024. All the results presented are evaluated without voting.

**Attention Score Regularization**    In section 3.2, we design two methods to compute the GDS. Before using $softmax$ function to obtain the attention map, we need to convert the GDS into attention score. Following the dot-product self-attention [36], we try to scale the attention score. We attempt to utilize three scaling settings on data manifold-based GSA, including "Norm.", "Inv." and "Neg." denote making the attention score normalization, inverse and negative, respectively. The results are illustrated in Fig. 3 (a). The accuracy results of the three settings are $92.5\%$, $92.3\%$, and $92.8\%$, which show that the redundant operation might impair the representation ability of GSA.

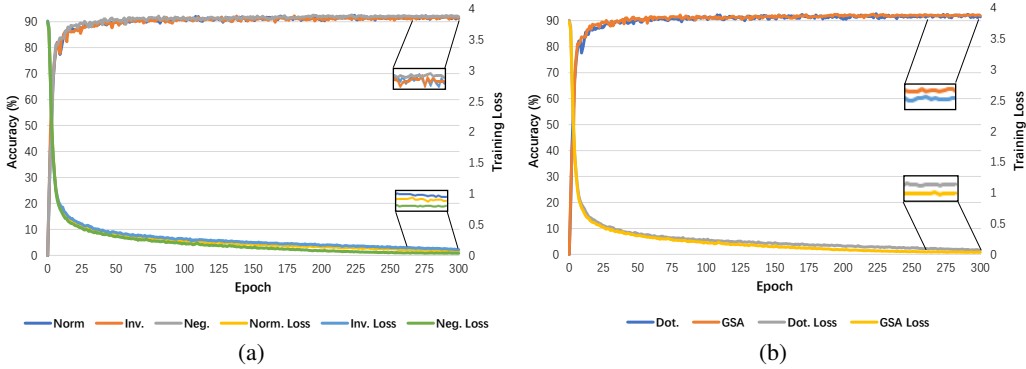

(a)                                        (b)

Figure 3: (a) **Ablation Study: Attention Score Regularization.** The validation accuracy curves and loss curves of three settings are presented. (b) **Ablation Study: Attention Types.** The validation accuracy curves and loss curves of dot-production self-attention and GSA are presented.

**Attention Types**    We compare several types of self-attention mechanisms in the Transformer block. The results are illustrated in Table 5 and some visual interpretations are provided in supplementary materials. Four conditions are evaluated as follows.

"Dot." denotes the standard dot product multiplication attention used in previous works. Given the input $X$, the operation can be described as:

$$Output = \text{Attention}(X) = \text{softmax}\left(\frac{W_Q(X)W_K^T(X)}{\sqrt{d_k}}\right)W_V(X). \tag{5}$$

"Dot.-less" denotes a dot product multiplication attention without linear transformations which can be described as:

$$Output = \text{Attention}(X) = \text{softmax}\left(\frac{XX^T}{\sqrt{d_k}}\right)W_V(X). \tag{6}$$

"Non." is a condition that we only apply a simple point-wise MLP to map the input $X$ without attention mechanism and the operation can be described as:

$$Output = W_V(X). \tag{7}$$

"GSA" denotes our geodesic self-attention and can be described as:

$$Output = \text{Attention}(X) = \text{softmax}\left(-\text{GDS}(X)\right)W_V(X). \tag{8}$$

The results fully demonstrate the superiority of our approach and the assumption that dot product multiplication in Euclidean space cannot accurately capture the internal non-Euclidean structure of point cloud objects. Further analyzing the results, we argue that the general linear transformation leads to reducing the ability of underlying geometric structure feature extraction of the point cloud,

which is fatal for point cloud analysis. Meanwhile, the redundant linear transformations map the input features to different high-dimensional spaces, aggravating such a phenomenon. To keep the geometric structure of the input features in the original space, Fei *et al.* [11] replace the conventional linear transformation with an orthogonal transformation and get a better result in 2D vision tasks, which helps interpret this phenomenon.

Table 5: **Ablation Study: attention types.** We report the accuracy (%) of classification on Model-Net40, the number of parameters and FLOPs of different types of self-attention.

| Methods | Dot. | Dot.-less | Non. | GSA-DM |
|---------|------|-----------|------|--------|
| Acc. | 91.7 | 92.1 | 92.3 | 92.8 |
| #Param. | 22.1M | 18.5M | 18.5M | 18.5M |
| FLOPs | 4.8G | 4.1G | 3.9G | 4.3G |

## 5  Conclusion

In the work, we assume that dot production multiplication in Euclidean space is not the optimal self-attention mechanism on capturing the complex non-Euclidean spatial structures of point clouds, and then propose a novel geodesic self-attention mechanism to solve this problem. The key insight behind the GSA is that the geodesic can obtain more accurate measurement of non-Euclidean structure of point clouds. Since the geodesic of point cloud is difficult to define and obtain, we propose two feasible methods to compute it. To verify our approach, we introduce the GSA into a standard Transformer structure, Point-GT, and it outperforms SOTA on different benchmarks.

## Acknowledgements

This work was supported by Natural Science Foundation of China (No. 61872147) and National Key Research and Development Program of China (No. 2018YFB2101300).

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
