# OpenReview forum: "Geodesic Self-Attention for 3D Point Clouds"
_NeurIPS.cc/2022/Conference — NeurIPS 2022 Accept_

### Official Review · Reviewer_2FWg · 2022-07-07

**Rating:** 6
**Confidence:** 3
**Soundness:** 4 excellent
**Presentation:** 4 excellent
**Contribution:** 3 good

**Summary:**

The article starts with the question of whether the dot product attention is the optimal operator and proposes the possibility of using geodesic distance for the calculation. The authors propose two ideas for the implementation of geodesic distance computation, the graph-based method and the data manifold-based method. The performance of the data manifold-based method is empirically verified on three tasks, which all achieves SOTA performance. For the graph-based method, the authors only give a feasibility check on the classification task.

**Questions:**

Please check the section strengths and weaknesses.

**Limitations:**

The method proposed in this article is an incremental improvement on existing methods and I have not seen any negative effects from it.

**Strengths And Weaknesses:**

Strengths:

On the presentation of the paper, the paper is very well written. In terms of content, for one problem, the authors try two different ideas to solve the problem of calculating the geodesic distance, and two different ways of thinking give the reader more insight.  On the experimental side, the authors organise empirical experiments that are able to demonstrate the feasibility of their graph-based approach and the effectiveness of their data manifold-based approach.

Weaknesses and Questions:

[1] The authors go to great lengths to explain the Graph-based method, and even implement it with GPU acceleration, but only demonstrate its feasibility in the experimental results. Given that the authors have implemented this part of the code, what are the results on the other two datasets? Also, the authors do not give any explanation for the difference in results between the two implementations, but simply pass over it. Some explanation of this part would be more enlightening to the reader.

[2] Line 324, when analyzing the result of different attention operator, the authors argue that "the general linear transformation causes scale ambiguity and destroys the geometric structure of the features, which is fatal for point cloud analysis." I'm a little bit confused at this part. The notion of scale ambiguity is a little bit vague to me. In the figure.5, if I understand correctly the GSA would means GSA-less which means it runs without the linear transformation. I was wondeing what's the result of GSA with the linear transformation. Will it performs worse?

[3] The authors do not mention anything about releasing the code, I'm a little concerned about its reproducibility without code.

Typo and minor things:

[1] Line 311, this is a figure refence issue.

[2] Fig.3 and Fig.3 are too close to each other.

---

> ### Author Response · Authors · 2022-08-01
> **Response to Reviewer 2FWg (Part 1 of Question 1)**
>
> We appreciate your careful review, positive feedback and constructive suggestions on this paper. The "Typo and minor things" have been corrected in the uploaded revision. The following responses might help address your questions and concerns about this paper:
>
> **1. Question 1.** "The authors go to great lengths to explain the Graph-based method, and even implement it with GPU acceleration, but only demonstrate its feasibility in the experimental results. Given that the authors have implemented this part of the code, what are the results on the other two datasets? Also, the authors do not give any explanation for the difference in results between the two implementations, but simply pass over it. Some explanation of this part would be more enlightening to the reader."
>
> **Answer to Question 1.** We highly appreciate your valuable advice. In the revision, we have illustrated the additional experimental results in **Section A.2 of the updated Supplementary Material**.
>
> (1) We use "(Graph-based)" and "(Data Manifold-based)"  to denote the proposed Graph-based GSA and Data Manifold-based GSA, respectively.
>
> (2) Specifically, we add more experiments to compare both geodesic metrics explicitly. The added results of **the classification task on ScanObjectNN** and **the part segmentation task on ShapeNetPart** are shown as follows:
>
> | Methods                        | OBJ-BG | OBJ-ONLY | PB-T50-RS |
> |--------------------------------|:------:|:--------:|:---------:|
> | Transformer                    |  79.9  |   80.6   |    77.2   |
> | Point-GT (Graph-based)         |    86.6    |     86.9     |    80.7   |
> | Point-GT (Data Manifold-based) |  87.6  |   87.6   |    81.7   |
>
> **Table.1 Additional classification results on ScanObjectNN.** We report the accuracy (%) of three different
> settings.
>
> | Methods                        | $mIoU_C$ | $mIoU_I$ | aero |  bag |  cap |  car | chair | earphone | guitar | knife | lamp | laptop | motor |  mug | pistol | rocket | skateboard | table |
> |--------------------------------|:--------:|:--------:|:----:|:----:|:----:|:----:|:-----:|:--------:|:------:|:-----:|:----:|:------:|:-----:|:----:|:------:|:------:|:----------:|:-----:|
> | Transformer |   83.42  |   85.1   | 82.9 | 85.4 | 87.7 | 78.8 |  90.5 |   80.8   |  91.1  |  87.7 | 85.3 |  95.6  |  73.9 | 94.9 |  83.5  |  61.2  |    74.9    |  80.6 |
> | Point-GT (Graph-based)         |   83.64  |   85.8   | 83.9 | 84.7 | 88.3 | 79.4 |  91.1 |   76.1   |  91.2  |  86.7 | 85.4 |  95.9  |  74.0 | 94.5 |  85.1  |  64.4  |    76.2    |  81.4 |
> | Point-GT (Data Manifold-based) |   84.15  |   85.8   | 84.3 | 84.5 | 88.3 | 80.9 |  91.4 |   78.1   |  92.1  |  88.5 | 85.3 |  95.9  |  77.1 | 95.1 |  84.7  |  63.3  |    75.6    |  81.4 |
>
> **Table.2 Additional part segmentation results on the ShapeNetPart dataset.** We report the $mIoU_C$ (\%), the $mIoU_I$ (\%), as well as the $IoU$ (\%) for each categories.
>
> In addition, the following details about the experiments of the graph-based GSA might be helpful for you:
> * During the experiments, the network should construct point cloud graph and compute the geodesic self-attention dynamically. As the size of the dynamic graph increases, the graph-based GSA’s consumption of computational resources grows steeply. Even if the graph-based GSA is accelerated by purposely designed CUDA arithmetic, such computational overhead is still unbearable.
> * In our experiments, when the input sampled points are increased 2 times, the dynamic graph size increases 4 times, and the computational time increases 8-12 times.
>
> All these experiments with more results have been added in Section A.2 of the updated Supplementary Material. More experiments are still on evaluated, and the results will be published as soon as possible.

---

> > ### Author Response · Authors · 2022-08-02
> > **Response to Reviewer 2FWg (Question 2-3)**
> >
> > **2. Question 2** "Line 324, when analyzing the result of different attention operator, the authors argue that "the general linear transformation causes scale ambiguity and destroys the geometric structure of the features, which is fatal for point cloud analysis." I'm a little bit confused at this part. The notion of scale ambiguity is a little bit vague to me. In the figure.5, if I understand correctly the GSA would means GSA-less which means it runs without the linear transformation. I was wondering what's the result of GSA with the linear transformation. Will it perform worse?"
> >
> > **Answer to Question 2.** We appreciate your valuable suggestion. We agree that "scale ambiguity" may confuse the readers. In the revision, the sentence "the general linear transformation causes scale ambiguity and destroys the geometric structure of the features, which is fatal for point cloud analysis." has been revised as "the general linear transformation leads to reducing the ability of underlying geometric structure feature extraction of the point cloud, which is fatal for point cloud analysis.".
> >
> > From a mathematical point of view, this assumption is that an unconstrained linear transformation may lead to distortion and deformation of the underlying geometric structure represented by the data.  **In Section A.3 and Figure 1 of updated Supplementary Material**, we have illustrated the attention heat maps about the several types of attention mentioned in the ablation experiments.
> > In Figure 1, the "Dot." denotes the standard dot-product self-attention, and the "Dot.-less" denotes the "Dot." without linear transformation. Both the graph-based GSA and the data manifold-based GSA are our proposed approaches. **As you mentioned, GSA can be understood as GSA-less, i.e., no linear transformation is used**.
> >
> > The result of the GSA with linear transformation evaluated on the classification task on the ModelNet40 dataset is 92.4% accuracy without voting, which performs worse than the GSA without linear transformation(92.8%).
> > It can be seen from Figure 1, some unrelated focus regions appear on the heat map of the "Dot." self-attention compared to the "Dot.-less", which confirms our assumption.
> >
> > **3. Question 3** "The authors do not mention anything about releasing the code, I'm a little concerned about its reproducibility without code."
> >
> > **Answer to Question 3.** Thank you for your recognition of our work. We will release the source codes on our homepage. The source codes are also will be uploaded to NeurIPS system associated with the paper if it could be accepted. Thank you.
> >
> > Thank you again for your time and provided such an excellent review!

---

> > ### Author Response · Authors · 2022-08-02
> > **Response to Reviewer 2FWg (Part 2 of Question 1):  explanation for the difference in results between two proposed methods**
> >
> > **Explanation**.
> >
> > As your suggestions, in the following, we now give more explanations for the difference in results between the two implementations, which have been shown in **Section A.2.1 of the updated Supplementary Material**. We give explanations from two aspects: methodology and visual interpretation experiments.
> >
> > **(1) Methodology.** Firstly, for the data that have complex structures with missing data points, e.g. ScanObjectNN dataset, Table.1 shows that graph-based GSA performs worse than manifold-based GSA. This may be because manifold-based GSA computes the better geodesic distances for the data that have complex structures.
> >
> > Secondly,  for the data that have simple structures with a small number of points, e.g. ShapeNetPart dataset, the computing of geodesic distances is easier. Graph-based GSA and manifold-based GSA have similar results, see Table 2.  The difference in the results possibly results from the incidental bias of the geodesic distances computing.
> >
> > Thirdly, manifold-based GSA mostly achieves better results in both datasets than Graph-based GSA. This might be because the shortest path algorithm of graph-based GSA is **non-differentiable**. In the paper, graph-based GSA first utilizes the KNN to compute the local neighborhood, and then the Floyd’s shortest path algorithm is used to estimate the geodesic distance of all pairs of the points (shown in lines 172-184 in the manuscript). This process is separated and non-differentiable in the whole model, as well as has complex computing. Although the powerful performance of PyTorch and Autograd makes the network work successfully, some imprecise gradient propagation may lead to differences in parameter accuracy and ultimately to worse results. As for the data manifold-based GSA, the projection manifold, Oblique Manifold, is differentiable which can avoid such a dilemma.
> > Overall, since the differentiality often affects the global optimization, this possibly leads to the difference in results.
> >
> > **(2) Visual interpretation.**  In **Section A.3 and Figure 1 of updated Supplementary Material**, we have illustrated the attention heat maps about the several types of attention mentioned in the ablation experiments. The visual interpretation results of the graph-based GSA show a worse ability to capture long-range dependencies, compared to the data manifold-based GSA.
> >
> >  In the work, the graph-based GSA is an alternative option that we construct for GSA. This provides more options for different applications. Additionally, constructing a better graph-based point cloud geodesic attention with GNN is also a good way to improve the graph-based GSA.

---

### Official Review · Reviewer_BX9m · 2022-07-08

**Rating:** 6
**Confidence:** 4
**Soundness:** 3 good
**Presentation:** 3 good
**Contribution:** 3 good

**Summary:**

This paper proposes to leverage geodesic metrics for computing the self-attention weights for point cloud transformer architectures. Two geodesic metrics are proposed and studied: one is graph-based and the other is manifold-based. The author conducted experiments on ModelNet and ScanObjectNN for object classification, ShapeNetPart for part segmentation, and ModelNet for few-shot learning. Comparing to baseline methods, the transformer networks equipped with the proposed geodesically defined self-attention modules demonstrate improved performance in all the tasks. Extensive ablation studies are also presented.

**Questions:**

- can you show results for both geodesic metrics explicitly in the table?
- can you compare to GeoNet or other methods that use geodesic distances as comparisons?

**Limitations:**

I don't have concerns about this.

**Strengths And Weaknesses:**

Strengths:
- the core idea of redefining the self-attention weights using the geodesic metrics over 3D point cloud data is novel, valid and interesting. To my best knowledge, the idea is quite novel.
- the experiments are very solid and convincing demonstrating better performance than baseline methods under different settings and for different tasks.

Weaknesses:
- the paper presents two ways for the geodesic metrics, but most tables in the experiment sections only report one "ours". It's much clearer to report both numbers using the two different metrics.
- can you compare to GeoNet or other methods that use geodesic distances as comparisons? None of the baselines the authors are comparing to is using any concept of geodesic distances. But there are papers, for example GeoNet, that are using geodesic metrics. Can you compare against these methods?

---

> ### Author Response · Authors · 2022-08-01
> **Response to Reviewer BX9m (Question 1)**
>
> We appreciate your careful review and feedback on this paper. The following responses might help address your questions about this paper:
>
> **1. Question 1.** " The paper presents two ways for the geodesic metrics, but most tables in the experiment sections only report one "ours". It's much clearer to report both numbers using the two different metrics. Can you show results for both geodesic metrics explicitly in the table?"
>
> **Answer to Question 1.** We highly appreciate your valuable advice. In the revision, we have illustrated the additional experimental results in **Section A.2 of the updated Supplementary Material**.
>
> (1) We use "(Graph-based)" and "(Data Manifold-based)"  to denote the proposed Graph-based GSA and Data Manifold-based GSA, respectively.
>
> (2) Specifically, we add more experiments to compare both geodesic metrics explicitly. The added results of **the classification task on ScanObjectNN** and **the part segmentation task on ShapeNetPart** are shown as follows:
>
> | Methods                        | OBJ-BG | OBJ-ONLY | PB-T50-RS |
> |--------------------------------|:------:|:--------:|:---------:|
> | Transformer                    |  79.9  |   80.6   |    77.2   |
> | Point-GT (Graph-based)         |    86.6    |     86.9     |    80.7   |
> | Point-GT (Data Manifold-based) |  87.6  |   87.6   |    81.7   |
>
> **Table.1 Additional classification results on ScanObjectNN.** We report the accuracy (%) of three different
> settings.
>
> | Methods                        | $mIoU_C$ | $mIoU_I$ | aero |  bag |  cap |  car | chair | earphone | guitar | knife | lamp | laptop | motor |  mug | pistol | rocket | skateboard | table |
> |--------------------------------|:--------:|:--------:|:----:|:----:|:----:|:----:|:-----:|:--------:|:------:|:-----:|:----:|:------:|:-----:|:----:|:------:|:------:|:----------:|:-----:|
> | Transformer |   83.42  |   85.1   | 82.9 | 85.4 | 87.7 | 78.8 |  90.5 |   80.8   |  91.1  |  87.7 | 85.3 |  95.6  |  73.9 | 94.9 |  83.5  |  61.2  |    74.9    |  80.6 |
> | Point-GT (Graph-based)         |   83.64  |   85.8   | 83.9 | 84.7 | 88.3 | 79.4 |  91.1 |   76.1   |  91.2  |  86.7 | 85.4 |  95.9  |  74.0 | 94.5 |  85.1  |  64.4  |    76.2    |  81.4 |
> | Point-GT (Data Manifold-based) |   84.15  |   85.8   | 84.3 | 84.5 | 88.3 | 80.9 |  91.4 |   78.1   |  92.1  |  88.5 | 85.3 |  95.9  |  77.1 | 95.1 |  84.7  |  63.3  |    75.6    |  81.4 |
>
> **Table.2 Additional part segmentation results on the ShapeNetPart dataset.** We report the $mIoU_C$ (\%), the $mIoU_I$ (\%), as well as the $IoU$ (\%) for each categories.
>
>
>
>
> All these experiments with more results have been added in Section A.2 of the **updated Supplementary Material**. More experiments are still on evaluated, and the results will be published as soon as possible.

---

> > ### Author Response · Authors · 2022-08-02
> > **Response to Reviewer BX9m (Question 2)**
> >
> > **2. Question 2.** "Can you compare to GeoNet or other methods that use geodesic distances as comparisons? None of the baselines the authors are comparing to is using any concept of geodesic distances. But there are papers, for example GeoNet, that are using geodesic metrics. Can you compare against these methods?"
> >
> > **Answer to Question 2.** We appreciate your suggestions very much. We performed a classification evaluation on the SHREC 15 dataset to compare with GeoNet, and the proposed graph-based GSA achieved a **95.83% accuracy which is better than GeoNet (94.67%)**. This will be added in the revision.
> >
> > Additionally, we would like to clarify that although GeoNet and our methods all introduce the geodesic metrics, they did not focus on the same problem:
> > * GeoNet computes the geodesic distance on the raw data before training the algorithm. So it is a method for describing the raw data which is mainly used for reconstructing the 3D data.
> > * Our method computes the geodesic on the feature maps in each Transformer layer. In detail, we can build an End-End network through the dynamically computed geodesic attention matrix.
> >
> > Since they often focus on different tasks, we did not give more relevant comparisons.
> >
> > We have **updated our Supplementary Material and illustrated the heat maps of several attention types for visual interpretation of GSA performance in Section A.3 and Table 1** which might better help you understand our contribution.
> >
> > Thanks again for your positive comments and constructive suggestions!

---

### Official Review · Reviewer_MUiF · 2022-07-11

**Rating:** 4
**Confidence:** 4
**Soundness:** 2 fair
**Presentation:** 2 fair
**Contribution:** 2 fair

**Summary:**

This paper propose the geodesic self-attention (GSA) module, which captures the long-range geometrical dependencies of point cloud objects by means of a Riemannian manifold on the metric, Riemannian manifold to capture the long-range geometrical dependencies of point cloud objects. dependencies of point cloud objects, and gives good results in terms of latency, parameters and nonlinear variability.

**Questions:**

1) I hope that the authors can add visual interpretation of the results of the Geodesic Self-Attention module used in their ablation experiments.
2) I don't see any relevant explanation of network loss in the paper and the supporting material. I hope the authors can give detailed loss settings for their experiments on downstream tasks.

In the current version, there are some issues as well. I look forward to the response by the authors. For now, I would recommend a borderline accept rating for the paper.

**Limitations:**

Yes, I don't see any potential negative social impact of this work.

**Strengths And Weaknesses:**

- Originality: The main idea of the proposed approach is to use the transformer with geodesic self-attention. To my limited knowledge about it, it lacks a little novelty, but the originality is reasonable.
- Quality：While the approach seems reasonable and the experimental results look promising, I have the following concerns(See Questions) about the paper.
- Clarity：This manuscript is clearly written.
- Significance：There are not many works in the field of point cloud feature learning by transformer, so I think this paper makes a positive contribution.

---

> ### Author Response · Authors · 2022-08-01
> **Response to Reviewer MUiF**
>
> We appreciate your careful comments and constructive suggestions. The followings are detailed responses to your questions/concerns:
>
> **1. Question 1.** "I hope that the authors can add visual interpretation of the results of the Geodesic Self-Attention module used in their ablation experiments."
>
>   **Answer to Question 1.** Thank you for this valuable suggestion. We agree that visual interpretations can help readers better understand the experiments' results. We have done the revision as follows:
> * We have now already updated the  **Supplementary Material**, in Section A.3 of which we show the heat maps of several self-attention mechanisms evaluated in the ablation experiments, as well as the experimental analysis. Please refer to **Figure 1 and Page 2-3 in Section A.3 of Supplementary Material**.
> * In **Figure 1**, the "Dot." denotes the standard dot-product self-attention, and the "Dot.-less" denotes the "Dot." without linear transformation. Both the graph-based GSA and the data manifold-based GSA are our proposed approaches. In the updated  **Supplementary Material**, **Figure 1** indicates the following results:
>      - The Dot-less self-attention ("Dot.-less" ) gives a better focus area than standard dot-product self-attention ("Dot."). This confirms our assumption that a general linear transformation may lead to reducing the ability of geometric structure feature extraction of the point cloud, i.e., a general linear transformation reduces the performance of the self-attention mechanism.
>     - The proposed GSA methods give more precise focus areas than the "Dot." and the "Dot.-less" methods. This confirms our motivation that similar Euclidean distances can not reveal the geodesic distances and geometric semantics. In short, the dot product multiplication in Euclidean space is insufficient to capture accurate internal non-Euclidean structures of point cloud objects.
>     - The graph-based GSA and the data manifold-based GSA present similar attention patterns, confirming our proposed methods' validity.
>
> **2. Question 2.** "I don't see any relevant explanation of network loss in the paper and the supporting material. I hope the authors can give detailed loss settings for their experiments on downstream tasks."
>
> **Answer to Question 2.** We agree that more detailed loss settings can help readers to reproduce the work. In the **updated Supplementary Material**,  we have done the following two revisions.
>
> (1) We illustrate the loss settings with the hyperparameters which are illustrated previously in **Section A.1 and Table 1** of Supplementary Material.
>
> (2) To further ensure a fair comparison of the experimental results, the settings of the loss function types in our network can be referred to previous works, such as Point-BERT and Point-MAE. Specifically, we set the loss as follows:
> * We chose the Cross-Entropy loss function for classification tasks (on ModelNet40 and ScanObjectNN) and few-shot learning tasks.
> * We used the Negative Log-Likelihood loss function for part segmentation tasks.
>
> We believe that choosing a commonly used loss function helps us evaluate the effectiveness of the proposed GSA.
>
>   **3. Your Conclusions.** "In the current version, there are some issues as well. I look forward to the response by the authors. For now, I would recommend a **borderline accept rating** for the paper."
>
>   **Reply to Your Conclusions.**  Thank you for your encouraging conclusions. In addition, we are a little confused that you concluded that "you would recommend a **borderline accept rating (5)** for the paper", but the temporary rating we obtained is **borderline reject rating (4)** in the above review system. Could you please consider revising your rating after reading our answers above? Thank you very much for your time.
>
>   **4. About the novelty.** Thanks again for your time and effort. To the best of our knowledge, we are the first to consider the problem that if a dot product multiplication attention is suitable for point cloud data, the first to introduce the geodesic concept into the attention mechanism, and the first to compute geodesic on feature maps of point clouds dynamically. Based on the above motivation, we proposed two novel and feasible geodesic self-attention methods. We hope these will enhance your recognition of our novelty.

---

> ### Author Response · Authors · 2022-08-09
> **A kind reminder.**
>
> Sorry to bother you again, we appreciate your effort in helping us make the paper stronger.
>
> You mentioned that "For now, I would recommend a **borderline accept rating (5)** for the paper." But the temporary rating we obtained is **borderline reject rating (4)** in the above review system.
> Could you please consider revising your rating after reading our responses? Could I give you a reminder now?

---

> ### Author Response · Authors · 2022-08-10
> **A kind reminder.**
>
> Sorry to bother you again, we appreciate your effort in helping us make the paper stronger.
>
> You mentioned that "For now, I would recommend a **borderline accept rating (5)** for the paper." But the temporary rating we obtained is a **borderline reject rating (4)** in the above review system. Could I give you a reminder now?

---

### Meta-Review · Area_Chair_5NPW · 2022-08-29

**Recommendation:** Accept
**Confidence:** Certain

**Metareview:**

The paper addresses an issue of existing self-attention module that is mainly designed for data on Euclidean domain; for those on non-Euclidean domains, e.g., those on Riemannian manifold, the paper proposes a Geodesic self-attention counterpart. Experiments on tasks of 3D classification and segmentation show the efficacy. All reviewers acknowledge the problem importance and contributions made in the paper, although a few concerns are raised, including additional ablation studies and comparisons with other methods using geodesic metrics.

In the rebuttal, the authors clearly respond and address the reviewers’ concerns. Acceptance is recommended. Congratulations!


**Award:**

No

---

### Decision · Program_Chairs · 2022-09-14

Accept